# Disentangling Neural Disjunctive Normal Form Models

**Kexin Gu Baugh**                                        KEXIN.GU17@IMPERIAL.AC.UK
*Imperial College London, UK*

**Vincent Perreault**                              VINCENT.PERREAULT@POLYMTL.CA
*Polytechnique Montréal, Canada*

**Matthew Baugh**                                MATTHEW.BAUGH17@IMPERIAL.AC.UK
*Imperial College London, UK*

**Luke Dickens**                                               L.DICKENS@UCL.AC.UK
*University College London, UK*

**Katsumi Inoue**                                                  INOUE@NII.AC.JP
*National Institute of Informatics, Japan*

**Alessandra Russo**                                       A.RUSSO@IMPERIAL.AC.UK
*Imperial College London, UK*

**Editors:** Leilani H. Gilpin, Eleonora Giunchiglia, Pascal Hitzler, and Emile van Krieken

## Abstract

Neural Disjunctive Normal Form (DNF) based models are powerful and interpretable approaches to neuro-symbolic learning and have shown promising results in classification and reinforcement learning settings without prior knowledge of the tasks. However, their performance is degraded by the thresholding of the post-training symbolic translation process. We show here that part of the performance degradation during translation is due to its failure to disentangle the learned knowledge represented in the form of the networks' weights. We address this issue by proposing a new disentanglement method; by splitting nodes that encode nested rules into smaller independent nodes, we are able to better preserve the models' performance. Through experiments on binary, multiclass, and multilabel classification tasks (including those requiring predicate invention), we demonstrate that our disentanglement method provides compact and interpretable logical representations for the neural DNF-based models, with performance closer to that of their pre-translation counterparts. Our code is available at `https://github.com/kittykg/disentangling-ndnf-classification`.

## 1. Introduction

Neuro-symbolic learning methods integrate neural models' learning capabilities with symbolic models' interpretability. These methods have expanded beyond differentiable inductive logic programming (Evans and Grefenstette, 2018; Glanois et al., 2022), showing success in reinforcement learning (Kimura et al., 2021; Delfosse et al., 2023; Baugh et al., 2025), transition systems (Gao et al., 2022; Phua and Inoue, 2024), and classification (Cingillioglu and Russo, 2021; Baugh et al., 2023). Their rule-based interpretations offer more faithful and reliable model explanations compared to post-hoc approaches (Rudin, 2019).

Many neuro-symbolic learning methods build upon inductive logic programming (ILP) (Muggleton and de Raedt, 1994; Law et al., 2018), requiring background knowledge for initial clauses (Sha et al., 2023), rule templates (Evans and Grefenstette, 2018; Shindo

et al., 2021; Glanois et al., 2022; Sen et al., 2022), mode biases (Cunnington et al., 2023b; Delfosse et al., 2023), or search space constraints (Shindo et al., 2023). While the ability to use background knowledge is a merit inherited from ILP, the hypothesis search space can be too large for the model to handle when prior knowledge is unavailable. And if the background knowledge injected by the user is incorrect, an optimal solution may be excluded from the search space. Additionally, these methods typically require predicate-based inputs, necessitating either manual definition (Delfosse et al., 2023) or extraction using pre-trained components (Kimura et al., 2021; Cunnington et al., 2023a; Shindo et al., 2023; Sha et al., 2023) when the inputs are not logical or structured. In contrast, neural methods such as MLPs can be trained without background knowledge, regardless of the input format, trading interpretability for flexibility. Decision trees are good alternatives to MLPs when interpretability is more important, but the trees can grow too large to be readable. The Optimal Sparse Decision Tree (OSDT) (Hu et al., 2019) addresses tree complexity but is limited to binary features and lacks support for multiclass classification. On the other hand, the recently developed neural DNF-based model (Cingillioglu and Russo, 2021; Baugh et al., 2023, 2025) represents a promising amalgamation of interpretability and flexibility, and operates without the need for task-specific background knowledge. Like MLPs, the neural DNF-based models are end-to-end differentiable, but under appropriate conditions can be translated to formal logic programmes. Unlike the mentioned neuro-symbolic approaches, the neural DNF-based models offer a broader space of possible rules and seamlessly integrate with other neural components (Cingillioglu and Russo, 2021; Baugh et al., 2025). However, performance can degrade when converted from neural to logical form, a critical step in providing interpretations for trained models.

In this paper, we make three major contributions. (1) We identify that the existing thresholding discretisation method fails to faithfully translate some neural units into logical form, causing a performance drop, and characterise such neural units as *entangled*. (2) We propose a disentanglement method to address this issue, better preserving model performance in the extracted symbolic representation. (3) We introduce a simple threshold-learning method for real-valued features. These learned bounds are of the form '*feature > learned threshold value*' and act as interpretable invented predicates. We demonstrate that our proposed disentanglement method drastically decreases the performance degradation when translating trained neural DNF-based models across binary, multiclass, and multilabel classification tasks (some of which require predicate invention).

## 2. Entangled Nodes in Neural DNF-based Models

A neural DNF model (Cingillioglu and Russo, 2021) is built up of *semi-symbolic nodes* that behave as soft conjunctions or disjunctions. Such a model is constructed with a layer of conjunctive nodes followed by a layer of disjunctive nodes. A semi-symbolic node with trainable weights $w_i$, $i = 1, \ldots, N$ and parameter $\delta$, under input $\mathbf{x}$ gives output:

$$\hat{y} = \tanh(f_{\mathbf{w}}(\mathbf{x})) = \tanh\left(\sum_{i=1}^{N} w_i x_i + \beta\right), \text{ with } \beta = \delta\left(\max_i |w_i| - \sum_{i=1}^{N} |w_i|\right) \quad (1)$$

where $x_i \in [-1, 1]$, with the extreme value 1 ($-1$) interpreted as $\top$ ($\bot$) and intermediate values interpreted as degrees of beliefs. Activation $\hat{y}$ is interpreted similarly, but cannot

take extreme values $\pm 1$. A bivalent interpretation of $\hat{y}$ treats $\hat{y} > 0$ ($\leq 0$) as $\top$ ($\bot$). $\delta$ induces behaviour similar to a conjunction (disjunction) when $\delta = 1$ ($-1$).[1]

We introduce the **soft-valued truth table** to describe the behaviour of a semi-symbolic node, as shown in Example 2.1.

**Definition 1 (Soft-valued Truth Table of Semi-symbolic Node)** *A soft-valued truth table captures the output behaviour of a semi-symbolic node with weights $\mathbf{w} \in \mathbb{R}^N$ over all possible bivalent inputs $\mathcal{X} = \{-1, 1\}^N$. Each row starts with a possible combination of input values $\mathbf{x} \in \mathcal{X}$, followed by the corresponding node's activation $\tanh(f_{\mathbf{w}}(\mathbf{x}))$ and bivalent interpretation $b = \top$ if $\tanh(f_{\mathbf{w}}(\mathbf{x})) > 0$ and $\bot$ otherwise.*

---

**Example 2.1: Soft-valued truth table**

Consider a trained conjunctive semi-symbolic node with weights $\mathbf{w} = [-6, -2, -2, 2, -6]$. Its bias is calculated as $\beta = \max |\mathbf{w}| - \sum |\mathbf{w}| = -12$. A soft-valued truth table of this node is shown below, with some rows with $b = \bot$ omitted for brevity.

| $x_1$ | $x_2$ | $x_3$ | $x_4$ | $x_5$ | Node activation $\tanh(\sum_{i=1}^{5} w_i x_i + \beta)$ | Bivalent Interpretation $b$ |
|---|---|---|---|---|---|---|
| 1 | 1 | 1 | 1 | 1 | -1.000 | $\bot$ |
| | | | | | ... | |
| -1 | 1 | -1 | 1 | -1 | 0.964 | $\top$ |
| -1 | -1 | 1 | 1 | -1 | 0.964 | $\top$ |
| -1 | -1 | -1 | 1 | -1 | 1.000 | $\top$ |
| -1 | -1 | -1 | -1 | -1 | 0.964 | $\top$ |

---

A neural DNF-based model can be translated to a symbolic representation via an automatic post-training process (Cingillioglu and Russo, 2021; Baugh et al., 2025). One key step in the post-training process is to convert the weights from a continuous range to a fixed-valued set $\{-6, 0, 6\}^N$ ($\pm 6$ to saturate tanh), so that the node can be translated into a bivalent logic representation with provable truth-value equivalence (Baugh et al., 2025). We call this the **discretisation** process. The discretisation method used in mentioned previous works is the thresholding method: it selects a value $\tau$ to filter out weights with small absolute values while maintaining the model's performance as much as possible:

$$g_{\text{threshold}}(\mathbf{w}; \tau) = \hat{\mathbf{w}}, \text{ with } \hat{w}_i = 6 \cdot \text{sign}(w_i) \cdot \mathbb{1}(|w_i| > \tau)$$

where $\mathbb{1}(\alpha) = 1$ if condition $\alpha$ is true and 0 otherwise and $0 \leq \tau \leq \max |\mathbf{w}|$. Thresholding is usually applied on the whole neural DNF-based model with a shared threshold $\tau$ across all conjunctive and disjunctive nodes (Cingillioglu and Russo, 2021; Baugh et al., 2023). Baugh et al. (2025) only thresholds the conjunctive nodes so that the conjunctive layer is translated into bivalent rules while translating the disjunctive layer into probabilistic rules. To encourage the weights to be in $\{-6, 0, 6\}^N$, Baugh et al. (2025) uses an auxiliary loss function $|w||6 - |w||$ during training. However, due to the flexibility of the neural architecture, a semi-symbolic node is more likely to learn an **entangled** representation that cannot be translated into a single logical formula.

Example 2.1 shows a conjunctive node with entangled weights. The discretisation that best preserves the node's behaviour uses $\tau = 0$, resulting in a discretised conjunctive node

---

1. See Appendix A for a more detailed background introduction.

with weights $[-6, -6, -6, 6, -6]$ and is translated into: c ← not $a_1$, not $a_2$, not $a_3$, $a_4$, not $a_5$. Although this is the best translation, it still alters the behaviour slightly: rule head c cannot be true when $a_4$ is not true, but it is possible for the original node's activation to be interpreted as true when $x_4 = -1$. This is because the thresholding method overemphasises the smaller magnitude weights $w_2, w_3$, and $w_4$. In fact, looking at the soft-valued truth table, the behaviour can be perfectly described by a set of three rules:

$$\mathfrak{L} = \left\{ \begin{array}{l} \text{c} \leftarrow \text{not } a_1, \text{ not } a_3, \text{ } a_4, \text{ not } a_5. \\ \text{c} \leftarrow \text{not } a_1, \text{ not } a_2, \text{ not } a_3, \text{ not } a_5. \\ \text{c} \leftarrow \text{not } a_1, \text{ not } a_2, \text{ } a_4, \text{ not } a_5. \end{array} \right\}.$$

By the lower-weighted terms only appearing in subsets of the rule bodies, it reflects the smaller influence of $x_2, x_3, x_4$ on the node's behaviour. This phenomenon often occurs in trained neural DNF-based models, despite having the auxiliary loss function $|w||6 - |w||$ to penalise such behaviour.

## 3. Disentanglement of Semi-symbolic Node

We propose a method that **disentangles** the weights of semi-symbolic nodes while preserving nodes' behaviour. The approach transforms a real-valued conjunctive node into multiple smaller discretised nodes that jointly represent the same behaviour as the original node's. The process consists of three steps (Figure 1): (1) split all conjunctive nodes into smaller disentangled and discretised conjunctive nodes; (2) replace the original conjunctive node with the split conjunctive nodes by connecting them to the disjunctive node with the same weight as the original node; and (3) if the model is not a neural DNF-MT model[2], apply the thresholding discretisation method on the disjunctive layer (either on 0 or sweeping through all possible values). We only disentangle conjunctive nodes while still applying thresholding to discretise the disjunctive layer because:

1. Split conjunctive nodes can be reconnected to the neural DNF-based model under logical equivalence without auxiliary predicates. For instance, if a conjunctive node $c$, connected to a disjunctive node $d$ with a positive weight, splits into $c_1, \ldots, c_k$, then $(c_1 \vee \ldots \vee c_k) \to d \equiv (c_1 \to d) \wedge \ldots \wedge (c_k \to d)$.

2. Step (1) of our method requires binary $\{-1, 1\}^N$ inputs. While conjunctive layer inputs are typically binary, disjunctive layer inputs cannot be guaranteed to be binary, even with auxiliary loss functions.

### 3.1. Disentangling a Conjunctive Node

To disentangle a conjunctive node, the goal is to find a collection of new conjunctive nodes that jointly encode its interpreted behaviour. To this end, we can make use of the following:

**Proposition 1** *Given a conjunctive node $c$ with $s$ inputs parameterised by $w \in \mathbb{R}^s$, there exists a collection of conjunctive nodes $\{c_i\}$ with associated (split) weights $\{\tilde{\mathbf{w}}_i\} = \mathcal{W}$, where each $\tilde{\mathbf{w}}_i$ is discretised, i.e. $\tilde{\mathbf{w}}_i \in \{-6, 0, 6\}^s$, such that under discrete input $\{-1, 1\}^s$ the disjunction of their bivalent interpretations is the same as that of $c$.*

---

2. Neural DNF-MT model proposed by Baugh et al. (2025) is designed to output probabilities.

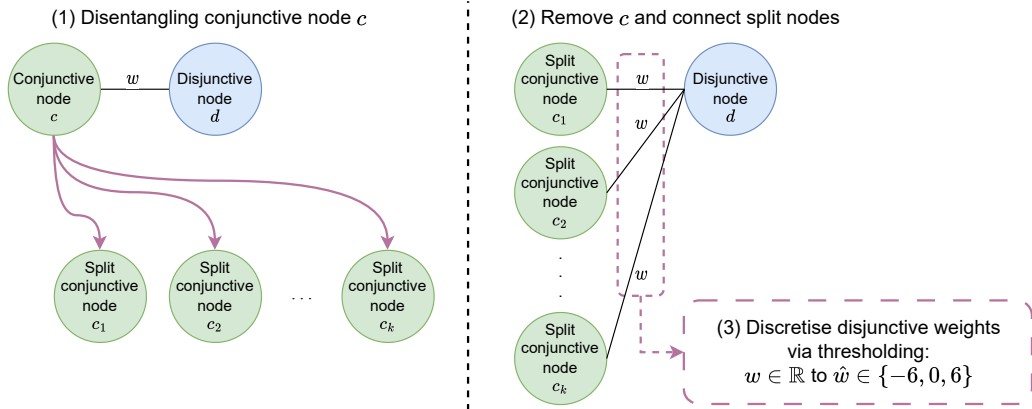

Figure 1: The three steps of the disentanglement method. Here $w$ is the weight connecting conjunctive node $c$ to disjunctive node $d$.

In this section we focus on disentangling a conjunctive node $c$ connected to a disjunctive node with *a positive weight*.[3]

Using Equation (1), the raw output of $c$ for input $\mathbf{x} \in \{-1, 1\}^s$ can be written as:

$$f_{\mathbf{w}}(\mathbf{x}) = \sum_{j \in \mathcal{J}} w_j x_j + \max_{j \in \mathcal{J}} |w_j| - \sum_{j \in \mathcal{J}} |w_j| \tag{2}$$

where the relevant set of indices of non-zero weights $\mathcal{J} = \{j \in \{1..s\} \,|\, w_j \neq 0\}$.

For any input $\mathbf{x}$ for which $c$ gives positive output there must be at least one split node $c_i$ with positive output, while the output of all split nodes for any other input must be less than or equal to zero. More formally, we define the set of positive inputs as $\mathcal{X}^+ = \{\mathbf{x} \in \{-1, 1\}^s | f_{\mathbf{w}}(\mathbf{x}) > 0\}$ and the set of negative inputs as $\mathcal{X}^- = \mathcal{X} \setminus \mathcal{X}^+$, and define the following two constraints on our collection of split nodes:

$$\forall \mathbf{x}^+ \in \mathcal{X}^+ . \exists i . f_{\tilde{\mathbf{w}}_i}(\mathbf{x}^+) > 0 \qquad \text{and} \qquad \forall \mathbf{x}^- \in \mathcal{X}^- . \forall i . f_{\tilde{\mathbf{w}}_i}(\mathbf{x}^-) \leq 0 \tag{3}$$

A *split weight set* $\mathcal{W} = \{\tilde{\mathbf{w}}_1, \tilde{\mathbf{w}}_2, \ldots\}$ for $c$ is the collection of weight tensors associated with the split nodes $\{c_1, c_2, \ldots\}$. A valid $\mathcal{W}$ can be constructed by associating each positive example $\mathbf{x}_i \in \mathcal{X}^+$ to a unique split node $c_i$ with weight tensor $\tilde{\mathbf{w}}_i$, where:

$$\forall j \in \{1 \ldots s\} . \tilde{w}_{i,j} = x_{i,j} \cdot \mathbb{1}\left(x_{i,j} = \text{sign}(w_j)\right) \cdot 6 = \begin{cases} 6 \cdot x_{i,j} & \text{if } x_{i,j} = \text{sign}(w_j) \\ 0 & \text{otherwise} \end{cases} \tag{4}$$

This split weight set $\mathcal{W}$ constructed under Equation (4) has the cardinality of $|\mathcal{X}^+|$, and the proof that it satisfies the constraints in Equation (3) is in Appendix B.1: the positive coverage is straightforward and we prove by contradiction for the negative coverage. $\mathcal{W}$ can be translated to a set of rules $\mathfrak{L}$, in the form of:

$$\mathfrak{L} = \left\{ c \leftarrow \bigwedge_{j \in \mathcal{J}_\ell^+} a_j, \bigwedge_{j \in \mathcal{J}_\ell^-} \text{not } a_j \ \middle| \ \ell \in \{1..|\mathcal{X}^+|\} \right\} \tag{5}$$

where $\mathcal{J}_\ell^+ = \{j \in \mathcal{J} | \tilde{w}_{\ell,j} = 6\}, \mathcal{J}_\ell^- = \{j \in \mathcal{J} | \tilde{w}_{\ell,j} = -6\}$.

---

3. For the negative weight case, see Appendix C.

### 3.2. Optimisation

In order to reduce the size of the split weight set, we can identify and remove redundant weight tensors. A weight tensor is redundant if its corresponding rule is logically subsumed by the rule of another weight tensor in the same split weight set. In this way, we can define a subsumption relation between two weight tensors in the split weight set as:

**Definition 2** *A weight tensor $\tilde{\mathbf{w}}_m$ **subsumes** $\tilde{\mathbf{w}}_n$ if:*

$$(\forall j \in \mathcal{J}.\tilde{w}_{m,j} \in \{0, \tilde{w}_{n,j}\}) \wedge (0 < |\{j \in \mathcal{J} : \tilde{w}_{m,j} \neq 0\}| < |\{j \in \mathcal{J} : \tilde{w}_{n,j} \neq 0\}|)$$

If $\tilde{\mathbf{w}}_n$, associated to $\mathbf{x}_n \in \mathcal{X}^+$, is subsumed by $\tilde{\mathbf{w}}_m$, $\tilde{\mathbf{w}}_m$ must also cover $\mathbf{x}_n$ (proof in Appendix B.2.1). This allows us to reduce $\mathcal{W}$ to a smaller yet equally valid split weight set $\mathcal{W}^*$ where no pair of weight tensors are subsumed by each other.

To compute $\mathcal{W}^*$, we introduce a three-step optimisation process:

1. **Compute the set of candidate indices of all small-magnitude weights in the original node that might not be essential to cover any positive examples.**
   The intuition is that weights with large absolute values contribute more to the output bivalent interpretation, and thus every large positive weight should be in all rules' positive sets of atoms $\mathcal{J}_\ell^+$ (Equation (5)) and every negative weight with large absolute value should be in all rules' negative sets of atoms $\mathcal{J}_\ell^-$. A weight $w_j$ is considered as having a large magnitude if $|w_j| \geq \max |\mathbf{w}|/2$, meaning $\bar{\mathcal{J}} = \{j \in \mathcal{J} \mid |w_j| < \max |\mathbf{w}|/2\}$ contains all the small-magnitude weights that might not be essential to cover any positive examples. The formal proof of this step, including how the threshold value $\max |\mathbf{w}|/2$ is computed, is available in Appendix B.2.

2. **Identify optimal combinations of candidate indices that can be excluded.**
   We define an exclusion set $\mathcal{E}_\ell$ of a weight tensor $\tilde{\mathbf{w}}_\ell$ as the set of indices of the non-zero elements in the original weight tensor $\mathbf{w}$ which were excluded by the disentanglement process when constructing $\tilde{\mathbf{w}}_\ell$, that is, $\{j \in \mathcal{J} | \tilde{w}_{\ell,j} = 0 \wedge w_j \neq 0\}$. By Remark 3, we can reformulate the definition: a set $\mathcal{E}$ is an exclusion set if and only if $\sum_{j \in \mathcal{E}} |w_j| < \max |\mathbf{w}|/2$. Since $\max |\mathbf{w}|/2$ is also the threshold value used to define $\bar{\mathcal{J}}$, an exclusion set $\mathcal{E}$ must be a subset of $\bar{\mathcal{J}}$, although a subset of $\bar{\mathcal{J}}$ is not necessarily an exclusion set. If $\tilde{\mathbf{w}}_\ell$ subsumes $\tilde{\mathbf{w}}_{\ell'}$, every zero element in $\tilde{\mathbf{w}}_{\ell'}$ must also be zero in $\tilde{\mathbf{w}}_\ell$, and $\tilde{\mathbf{w}}_\ell$ must have more zero elements than $\tilde{\mathbf{w}}_{\ell'}$, thus $\mathcal{E}_\ell \supset \mathcal{E}_{\ell'}$. We can therefore transform the original optimisation problem over finding $\mathcal{W}^*$ to finding $E^* = \{\mathcal{E}_1^*, \ldots, \mathcal{E}_\ell^*, \ldots, \mathcal{E}_L^*\}$ where no pairs of elements are subset of each other. To find $E^*$, we can perform a search by considering possible candidate $\hat{\mathcal{J}} \subset \bar{\mathcal{J}}$ and verifying it is an exclusion set using $\sum_{j \in \hat{\mathcal{J}}} |w_j| < \max |\mathbf{w}|/2$. Full details of this search are in Algorithm 1.

3. **Convert exclusion sets to weight tensors.** For each exclusion set $\mathcal{E}_\ell^* \in E^*$, we create a new weight tensor $\tilde{\mathbf{w}}_\ell$ where its $j$-th element is:

$$\tilde{w}_{\ell,j} = \begin{cases} 6 \cdot \text{sign}(w_j) & \text{if } j \in \mathcal{J} \setminus \mathcal{E}_\ell \\ 0 & \text{otherwise} \end{cases}$$

Thus, we compute $\mathcal{W}^* = \{\tilde{\mathbf{w}}_1, \ldots, \tilde{\mathbf{w}}_L\}, L \leq |\mathcal{X}^+|$, where no pair of weight tensors are subsumed by each other.

## 4. Predicate Invention

Neural DNF-based models require predicate invention to handle non-bivalent inputs. Cingilli-oglu and Russo (2021) and Baugh et al. (2025) use neural networks (Figure 2(b)) for this purpose, extracting higher-level features as predicates from low-level unstructured inputs. However, these neural predicates are not inherently interpretable, and the attempt of Baugh et al. (2025) to improve the interpretability through an ASP optimisation process does not guarantee a solution.

We propose an inherently interpretable approach called **threshold-learning predicate invention** (Figure 2(c)). For each real-valued input feature $x_i$ we learn $m$ threshold values $\{t_{i,1}, \ldots t_{i,m}\}$, which are each used to create a predicate $p_{i,j} = \tanh((x_i - t_{i,j})/T)$ where $T \in [0.1, 1]$ is a temperature parameter. The invented predicate $p_{i,j}$ can be interpreted as a bivalent predicate representing the inequality $x_i > t_{i,j}$ if $p_{i,j} > 0$. We schedule the temperature $T$ to decrease throughout training to push $p_{i,j}$ away from 0, thus amplifying the degree of the belief of the invented predicate. Our threshold-learning predicate invention does not need post-training methods to be interpretable and guarantees a solution.

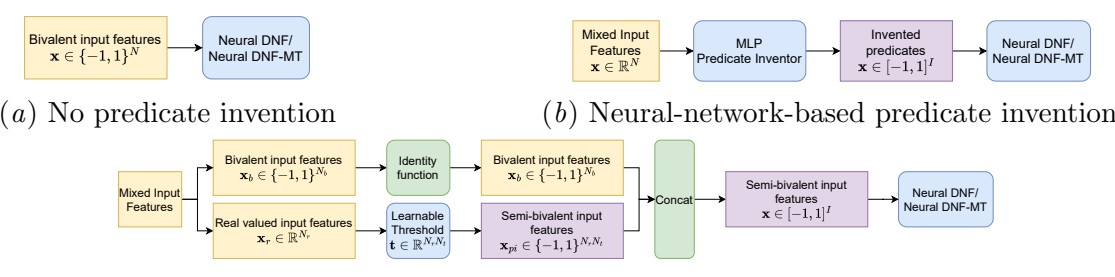

$(a)$ No predicate invention $\qquad\qquad$ $(b)$ Neural-network-based predicate invention

$(c)$ Threshold-learning predicate invention

Figure 2: Architectures for various settings, where blue rounded boxes have learnable parameters. Neural DNF is used in binary or multilabel tasks, while neural DNF-MT in multiclass tasks. 2(a) is the basic setup for bivalent inputs. 2(b) and 2(c) show how predicate invention is performed with MLPs and threshold learning.

## 5. Evaluation

### 5.1. Experiments

**Set up.** We train neural DNF-based models across *binary*, *multiclass* and *multilabel* classification tasks, including tasks with real-valued features that require *predicate inventions*:

- Binary: BCC[†] (Patrício et al., 2018), Monk (Wnek, 1993), Mushroom (Guide, 1981) and CDC[†] (Teboul, 2021).
- Multiclass: Car (Bohanec, 1988) and Covertype[‡] (Blackard, 1998).
- Multilabel: ARA (Chaos et al., 2006), Budding (Li et al., 2004), Fission (Davidich and Bornholdt, 2008), and MAM (Fauré et al., 2006).[4]

---

4. These are Boolean Network (Kauffman, 1969) datasets with ground-truth logic programs. Boolean Network models the state transitions of genes from time $t$ to $t + 1$, and we treat the learning of such a dynamic system as a multilabel classification problem, where the labels are the gene states at time $t+1$.

Datasets with † and ‡ require predicate inventions, and we use threshold-learning predicate invention for † and MLP predicate invention for ‡. All experiments are repeated at least 5 times. Further experiment details can be found in Appendix D.2.

**Results.** The first rows in Table 1 and 2 show that neural DNF-based models achieve competitive F1 scores compared to standard ML approaches that also do not rely on task-specific background knowledge, with neural DNF-based models performing the best in 7 of the 10 datasets. The neural DNF-based models match the level of expressiveness while enabling end-to-end training with predicate invention for real-valued inputs.

Table 1: F1 scores (mean ± ste) for binary and multilabel classification tasks.

| | Binary Classification | | | | Multilabel Classification | | | |
|---|---|---|---|---|---|---|---|---|
| Model | Monk | BCC† | Mushroom | CDC† | ARA | Budding | Fission | MAM |
| NDNF | $1.000 \pm 0.000$ | $0.718 \pm 0.072$ | $1.000 \pm 0.000$ | $0.882 \pm 0.001$ | $1.000 \pm 0.000$ | $1.000 \pm 0.000$ | $1.000 \pm 0.000$ | $1.000 \pm 0.000$ |
| NDNF: ASP | $0.992 \pm 0.008$ | $0.765 \pm 0.033$ | $0.971 \pm 0.016$ | $0.861 \pm 0.003$ | $0.993 \pm 0.002$ | $0.993 \pm 0.003$ | $0.987 \pm 0.003$ | $0.994 \pm 0.003$ |
| MLP | $1.000 \pm 0.000$ | $0.725 \pm 0.025$ | $0.999 \pm 0.000$ | $0.896 \pm 0.000$ | $1.000 \pm 0.000$ | $1.000 \pm 0.000$ | $1.000 \pm 0.000$ | $1.000 \pm 0.000$ |
| Logistic Regression | $0.707 \pm 0.006$ | $0.796 \pm 0.020$ | $1.000 \pm 0.000$ | $0.873 \pm 0.000$ | NA | NA | NA | NA |
| SVM | $1.000 \pm 0.000$ | $0.762 \pm 0.029$ | $1.000 \pm 0.000$ | $0.894 \pm 0.000$ | NA | NA | NA | NA |
| Random Forest | $0.984 \pm 0.005$ | $0.753 \pm 0.012$ | $1.000 \pm 0.000$ | $0.889 \pm 0.000$ | $1.000 \pm 0.000$ | $0.996 \pm 0.000$ | $0.992 \pm 0.001$ | $0.998 \pm 0.001$ |
| Decision Tree | $0.877 \pm 0.027$ | $0.766 \pm 0.030$ | $1.000 \pm 0.000$ | $0.871 \pm 0.002$ | $1.000 \pm 0.000$ | $0.972 \pm 0.001$ | $0.954 \pm 0.002$ | $0.950 \pm 0.002$ |
| OSDT | $1.000 \pm 0.000$ | NA | $0.893 \pm 0.000$ | NA | NA | NA | NA | NA |

The second rows of Table 1 and 2 present the performance of the novel optimised logical interpretation of our models as logic programs (NDNF: ASP and NDNF: LI resp.), which in most cases leads to a minor performance drop compared to the neural DNF-based models but at the the benefit of interpretability. Table 3 shows that our new interpretation method has a much smaller reduction in performance (right column) compared with the pre-existing threshold method (left column) in all but one dataset (Covertype). This exception may relate to the lack of faithfulness guarantee for conjunctive node disentanglement when using predicate invention, along with a higher task complexity.

Table 2: F1 scores (mean ± ste) for multiclass tasks. 'NDNF-MT: LI' is the logical interpretation of neural DNF-MT models.

| Model | Car | Covertype‡ |
|---|---|---|
| NDNF-MT | $0.963 \pm 0.001$ | $0.859 \pm 0.003$ |
| NDNF-MT: LI | $0.935 \pm 0.005$ | $0.823 \pm 0.003$ |
| MLP | $0.869 \pm 0.005$ | $0.828 \pm 0.001$ |
| Log. Reg. | $0.891 \pm 0.003$ | $0.758 \pm 0.000$ |
| SVM | $0.937 \pm 0.004$ | $0.716 \pm 0.001$ |
| Random Forest | $0.954 \pm 0.002$ | $0.942 \pm 0.000$ |
| Decision Tree | $0.959 \pm 0.004$ | $0.914 \pm 0.002$ |

Table 3: F1 score drops after thresholding $(f1_t - f1_{thresh})$ and disentanglement $(f1_t - f1_{disent})$ discretisation.

| | Dataset | $f1_{train} - f1_{thresh}$ | $f1_{train} - f1_{disent}$ |
|---|---|---|---|
| Bin. | Monk | $0.076 \pm 0.027$ | $\mathbf{0.008 \pm 0.008}$ |
| | BCC† | $0.050 \pm 0.038$ | $\mathbf{0.003 \pm 0.022}$ |
| | Mush | $0.316 \pm 0.099$ | $\mathbf{0.029 \pm 0.016}$ |
| | CDC† | $0.031 \pm 0.007$ | $\mathbf{0.021 \pm 0.003}$ |
| MC. | Car | $0.122 \pm 0.012$ | $\mathbf{0.027 \pm 0.005}$ |
| | Cover‡ | $\mathbf{0.036 \pm 0.003}$ | $0.043 \pm 0.004$ |
| ML. | ARA | $0.085 \pm 0.009$ | $\mathbf{0.007 \pm 0.002}$ |
| | Budd | $0.159 \pm 0.004$ | $\mathbf{0.007 \pm 0.003}$ |
| | Fission | $0.141 \pm 0.007$ | $\mathbf{0.013 \pm 0.003}$ |
| | MAM | $0.045 \pm 0.004$ | $\mathbf{0.005 \pm 0.003}$ |

To assess the compactness of the logical interpretations induced by our disentangled neural DNF-based models we compare the length of the rules of its ASP program (full ASP programs for binary/multilabel tasks, ASP programs for conjunctive layers in multiclass

task) with the tree sizes of decision trees and OSDTs trained on the same datasets[5]. For a decision tree's decision path, each node in the path can be seen as a condition clause of an if statement, thus making the decision tree's depth an equivalent metric to ASP rule length. The number of branches indicates the number of different decision paths, making it equivalent to the number of rules in an ASP program. Table 4 shows our models' interpretations are consistently more compact than those of decision trees, as shown by them having fewer rules and both the average and maximum rule length being lower. Comparing to OSDT's, we have comparable compactness (we are more compact for the Monk dataset while OSDT is more compact for the Mushroom dataset) whilst achieving superior performance.

Table 4: Comparing compactness of neural DNF-based models' logic interpretations vs. decision trees', i.e. max. rule length/tree depth, avg. rule length/tree depth, and no. rules/branches. The cell marked with * refers to OSDT's performance.

| | Dataset | Max. Rule Length/Max Depth NDNF/ NDNF-MT: ASP | DT/ *OSDT | Avg. Rule Length/Avg Depth NDNF/ NDNF-MT: ASP | DT/ *OSDT | No. Rules/No. Branches NDNF/ NDNF-MT: ASP | DT/ *OSDT |
|---|---|---|---|---|---|---|---|
| Bin. | Monk | $2.778 \pm 0.148$ | $10.000 \pm 0.283$ *$4.000 \pm 0.000$ | $2.042 \pm 0.053$ | $7.842 \pm 0.098$ *$3.286 \pm 0.000$ | $4.000 \pm 0.000$ | $49.000 \pm 7.071$ *$7.000 \pm 0.000$ |
| | BCC[†] | $3.200 \pm 0.179$ | $6.600 \pm 0.219$ | $2.267 \pm 0.121$ | $4.833 \pm 0.062$ | $5.000 \pm 0.566$ | $18.200 \pm 0.657$ |
| | Mush | $4.250 \pm 0.225$ | $7.400 \pm 0.358$ *$2.400 \pm 0.219$ | $2.982 \pm 0.122$ | $4.907 \pm 0.132$ *$1.900 \pm 0.128$ | $3.750 \pm 0.242$ | $18.000 \pm 1.095$ *$3.400 \pm 0.219$ |
| | CDC[†] | $4.800 \pm 0.276$ | $40.600 \pm 0.607$ | $2.021 \pm 0.076$ | $21.804 \pm 0.021$ | $27.600 \pm 0.851$ | $31394 \pm 257.045$ |
| MC. | Car | $8.500 \pm 0.153$ | $13.000 \pm 0.283$ | $4.432 \pm 0.075$ | $9.462 \pm 0.060$ | $25.688 \pm 0.943$ | $109.200 \pm 1.927$ |
| | Cov[‡] | $4.300 \pm 0.202$ | $25.200 \pm 0.657$ | $1.778 \pm 0.041$ | $14.541 \pm 0.047$ | $38.900 \pm 1.090$ | $868.400 \pm 8.083$ |
| ML. | ARA | $3.900 \pm 0.095$ | $13.000 \pm 0.000$ | $2.050 \pm 0.024$ | $9.724 \pm 0.006$ | $26.700 \pm 0.318$ | $581.800 \pm 2.125$ |
| | Bud | $4.000 \pm 0.000$ | $12.000 \pm 0.000$ | $3.126 \pm 0.036$ | $10.937 \pm 0.007$ | $48.400 \pm 1.070$ | $1521.300 \pm 6.406$ |
| | Fis | $6.000 \pm 0.000$ | $10.000 \pm 0.000$ | $2.852 \pm 0.020$ | $9.308 \pm 0.018$ | $21.800 \pm 0.276$ | $392.500 \pm 4.401$ |
| | MAM | $4.000 \pm 0.000$ | $10.000 \pm 0.000$ | $2.645 \pm 0.014$ | $0.897 \pm 0.018$ | $21.500 \pm 0.212$ | $424.200 \pm 5.261$ |

To summarise, the neural DNF-based models demonstrate comparable performance to other methods that also do not rely on background knowledge and train well together with end-to-end predicate invention, while also providing compact logical representations at a minimal performance cost. In particular, our new disentanglement approach reduces such performance loss when discretising the neural DNF-based models compared to the previous thresholding method.

## 5.2. Discussion

**Limitations when Disentangling Disjunctive Layers.** We observe entanglements involving both conjunctive and disjunctive nodes lead to imperfect disentanglement: Example 5.1 shows a case where disjunctive nodes assign weights to utilise small-magnitude activations of conjunctive nodes to match the target. As discussed in Section 3, the disentanglement requires node inputs to be exactly $\{-1, 1\}^N$. Despite using an auxiliary loss to encourage $\pm 1$ conjunctive activations, this constraint is not guaranteed for the disjunctive nodes. This prevents applying the same disentanglement method to the disjunctive layer, and we find that splitting disjunctive nodes performs worse than simply thresholding in our experiments. Future works can investigate how to avoid these intermediate-valued cases.

---

5. We focus only on decision trees and OSDTs as direct competitors of neural DNF-based models, as they also provide rule-based decision processes, whereas other used ML baselines lack such explicit interpretability. More discussion on model interpretability can be found in Appendix D.1.

**Scalability of Disentanglement.** The disentanglement method's runtime grows exponentially with the number of inputs used in a node, as shown in Figure 3, since it may check all possible weight combinations that trigger the node to fire for the worst-case scenario. This makes it less scalable than the thresholding discretisation method, which scales linearly with weight magnitudes. However, this limitation only affects model interpretation, not learning capability, and can be mitigated through model sparsification.

---

**Example 5.1: Nested Entanglement**

Consider two conjunctive nodes with weights $\mathbf{w}_{c1} = [-1.33, 0, 0, 1.01, -1.44]$ and $\mathbf{w}_{c2} = [-2.02, -0.79, -0.79, 0.71, -1.52]$ respectively; and two disjunctive nodes connected to those conjunctive nodes, with weights $\mathbf{w}_{d1} = [3.43, 1.28]$ and $\mathbf{w}_{d2} = [0, 3.56]$ respectively. There is a nested entanglement among all the nodes here: $c1$ subsumes $c2$, and $d1$ uses a combination of $c1$ and $c2$ to match the target label. The disjunctive nodes' bivalent interpretations change depending on whether the input activations from the conjunctive nodes are soft-valued or strictly $\pm 1$, even when the interpretations of the conjunctive nodes' activations are the same (differences marked in red in the table below).

| a1 | a2 | a3 | a4 | a5 | c1 | c2 | d1 | d2 |
|----|----|----|----|----|------|------|------|------|
| -1 | -1 | -1 | -1 | -1 | -0.52/⊥ | 0.54/⊤ | 0.17/⊤ | 1.00/⊤ |
|    |    | -  |    |    | -1/⊥ | 1/⊤ | -0.70/⊥ | 1.00/⊥ |
| -1 | -1 | -1 | 1 | 1 | -0.89/⊥ | -0.77/⊥ | -0.99/⊥ | 0.67/⊤ |
|    |    | -  |    |    | -1/⊥ | -1/⊥ | -1.00/⊥ | 0.00/⊥ |

---

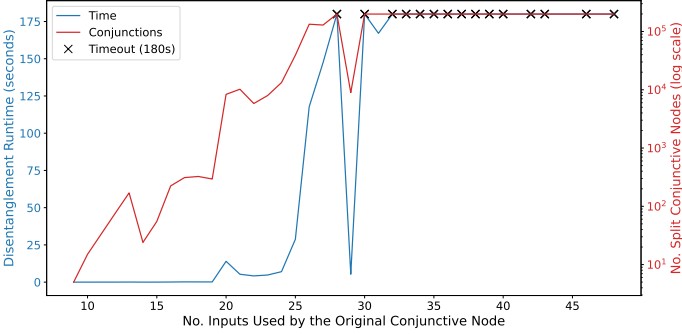

Figure 3: Time taken (blue) to disentangle the conjunctive nodes with different numbers of inputs used, and the number of split conjunctions (red) after disentanglement, in a trained neural DNF model. A max runtime of 180s is set for disentanglement.

## 6. Conclusion

We propose a disentanglement discretisation method that enables more faithful translation of neural DNF-based models into bivalent logic representations. Additionally, we introduce an interpretable-by-design predicate invention method that learns threshold values end-to-end, effectively converting real-valued features into inequalities as boolean predicates. Our experiments across various real-world classification tasks demonstrate that the disentanglement method provides compact and interpretable logical representations for the neural DNF-based models without requiring task-specific background knowledge, while improving performance over the existing thresholding method. We suggest that further research into training regimes may help optimise the trade-off between performance and interpretability for neural DNF-based models.

## Acknowledgments

This work is supported in part by DEVCOM Army Research Lab under W911NF2220243; EPSRC projects EP/X040518/1 and EP/Y037421/1; JST CREST Grant Number JP-MJCR22D3; JSPS KAKENHI Grant Number JP25K03190; and the NII Internship Program. We acknowledge also the project NIHR i4i: Artificial Intelligence to support cancer early diagnosis in general practice (NIHR207533). The views expressed in this paper are those of the author(s) and not necessarily those of the NIHR or the Department of Health and Social Care.

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

## Appendix A. Semi-symbolic Node and Neural DNF-based Model

A neural Disjunctive Normal Form model Cingillioglu and Russo (2021) is a neural model with nodes that can behave as a semi-symbolic conjunction or disjunction. The formalisation of a semi-symbolic node is listed in Equation (1), and the neural DNF model itself is constructed with a layer of conjunctive nodes followed by a layer of disjunctive nodes. The inputs to the semi-symbolic node are strictly bounded in $[-1, 1]$, where the extreme value $1\ (-1)$ is interpreted as $\top\ (\bot)$ and the intermediate values represent weaker strengths of belief. The node activation $\hat{y}$ can be interpreted similarly but cannot take the extreme values $\pm 1$. A bivalent interpretation of $\hat{y}$ is to treat $\hat{y} > 0\ (\leq 0)$ as $\top\ (\bot)$. $\delta$ induces behaviour similar to a conjunction (disjunction) when $\delta = 1\ (= -1)$. During training, the absolute values of $\delta$ in both layers gradually increase to 1 (controlled by a scheduler) for better learning. Cingillioglu and Russo (2021) shows how a neural DNF model can be used in binary classification. Baugh et al. (2023) proposes an extended model called the neural DNF-EO for multi-class classifications. It has a non-trainable constraint layer after the disjunctive layer to ensure a logically mutually exclusive output. Baugh et al. (2025) proposes a neural DNF-MT model. The tanh activation in the final disjunctive layer is replaced with a mutex-tanh activation function, so that the model can represent mutually-exclusive probability distributions as its outputs for reinforcement learning. Baugh et al. (2025) also shows how end-to-end predicate-invention can be performed with the model. All these methods interpret trained neural DNF models with 'deterministic' behaviour as logical rules represented in Answer Set Programming (Lifschitz, 2019). Neural DNF-MT model also supports probabilistic interpretation in ProbLog (De Raedt et al., 2007) for its disjunctive layer but its conjunctive layer is still translated to deterministic rules.

---

**Example A.1: Bivalent logic translation**

A symbolisation function $\sigma$ maps names of inputs ($x_i$) or nodes' bivalent interpretations ($b_{ci}/b_{di}$ for conj./disj. nodes) to symbolic predicate names: e.g. $\sigma(x_i) = a_i$, $\sigma(b_{ci}) = c_i$ and $\sigma(b_{di}) = d_i$. Under this $\sigma$, a conjunctive semi-symbolic node with weights $[6, 0, 0, -6]$ is translated into $c \leftarrow a_1, \text{not } a_4$. Similarly, a disjunctive semi-symbolic node with conjunctive nodes as inputs and weights $[0, -6, 6]$ is translated into $\{d \leftarrow \text{not } c_2.\ d \leftarrow c_3.\}$.

---

## Appendix B. Disentanglement of Positively Connected Conjunction

This section follows the setting in Section 3, where we want to disentangle a conjunctive node positively connected to a disjunctive node, i.e. the weight connecting them is positive.

Recall the goal split weight set $\mathcal{W} = \{\tilde{\mathbf{w}}_1, \tilde{\mathbf{w}}_2, \dots, \tilde{\mathbf{w}}_L\}$ in Section 3.1. The weights must have the form:

$$\forall \ell \in \{1..L\}, i \in \{1..s\}. \begin{cases} \tilde{w}_{\ell,i} \in \{-6, 0, 6\} & \text{if } i \in \mathcal{J} \\ \tilde{w}_{\ell,i} = 0 & \text{otherwise} \end{cases}. \tag{6}$$

Constraints 3 can be rewritten as:

$$\forall \mathbf{x} \in \mathcal{X}^+ . \exists \ell \left[ \sum_{j \in \mathcal{J}} \tilde{w}_{\ell,j} x_j + \left( 6 - 6 \sum_{j \in \mathcal{J}} |\tilde{w}_{\ell,j}| \right) = 6 \right] \tag{7}$$

$$\forall \mathbf{x} \in \mathcal{X}^- . \neg \exists \ell \left[ \sum_{j \in \mathcal{J}} \tilde{w}_{\ell,j} x_j + \left( 6 - 6 \sum_{j \in \mathcal{J}} |\tilde{w}_{\ell,j}| \right) > -6 \right] \tag{8}$$

This means that for every positive example $(\mathbf{x}^+)$ at least one rule is activated, and for every negative example $(\mathbf{x}^-)$ none of the rules are activated.

### B.1. Proof of Coverage of Split

Recall the splitting method in Section 3.1. Here we prove that the set $\mathcal{W} = \{\tilde{\mathbf{w}}_1, \ldots, \tilde{\mathbf{w}}_{|\mathcal{X}^+|}\}$ constructed under Equation (4) satisfies Conditions (7) and (8), i.e. all positive examples are covered and no negative example is covered.

**Proof Positive Coverage**

The positive coverage is straightforward. For any positive example $\mathbf{x}_i \in \mathcal{X}^+$, take $\tilde{\mathbf{w}}_i$ associated to $\mathbf{x}_i$. By Equation (4), we have:

$$
\begin{aligned}
&\sum_{j \in \mathcal{J}} \tilde{w}_{i,j} x_{i,j} + \left( 6 - \sum_{j \in \mathcal{J}} |\tilde{w}_{i,j}| \right) \\
&= 6 \sum_{j \in \mathcal{J}} \mathbb{1}\left( x_{i,j} = \text{sign}(w_j) \right) + \left( 6 - 6 \sum_{j \in \mathcal{J}} \mathbb{1}\left( x_{i,j} = \text{sign}(w_j) \right) \right) \\
&= 6
\end{aligned}
\tag{9}
$$

The set $\{\tilde{\mathbf{w}}_1, \ldots, \tilde{\mathbf{w}}_{|\mathcal{X}^+|}\}$ will cover all positive examples $\mathbf{x}_i \in \mathcal{X}^+$. ∎

**Proof Negative Coverage**

We will prove that the set $\{\tilde{\mathbf{w}}_1, \ldots, \tilde{\mathbf{w}}_{|\mathcal{X}^+|}\}$ constructed under Equation (4) does not cover any of the negative examples by contradiction.

Assume that $\mathbf{x}_m \in \mathcal{X}^-$ is covered by $\tilde{\mathbf{w}}_i$, which is associated to positive example $\mathbf{x}_i \in \mathcal{X}^+$. Then the raw output of $\tilde{\mathbf{w}}_i$ over $\mathbf{x}_m$ should be 6 to cover $\mathbf{x}_m$:

$$f_{\mathbf{w}_i}(\mathbf{x}_m) = 6$$

$$
\begin{aligned}
&\sum_{j \in \mathcal{J}} \tilde{w}_{i,j} x_{m,j} + \left( 6 - \sum_{j \in \mathcal{J}} |\tilde{w}_{i,j}| \right) = 6 \\
\Rightarrow &\sum_{j \in \mathcal{J}} \tilde{w}_{i,j} x_{m,j} = \sum_{j \in \mathcal{J}} |\tilde{w}_{i,j}|
\end{aligned}
\tag{10}
$$

Substitute the definition of $\tilde{w}_{i,j}$ in Equation (4) into Equation (10), and we have:

$$\sum_{j \in \mathcal{J}} x_{i,j} \cdot \mathbb{1}\left( x_{i,j} = \text{sign}(w_j) \right) \cdot x_{m,j} = \sum_{j \in \mathcal{J}} \mathbb{1}\left( x_{i,j} = \text{sign}(w_j) \right) \tag{11}$$

When $x_{i,j} \neq \text{sign}(w_j)$, $\mathbb{1}(x_{i,j} = \text{sign}(w_j)) = 0$, and will not contribute to the summation term in the L.H.S of Equation (11). Thus, for Equation (11) to hold, this has to hold:

$$\forall j \in \mathcal{J}. \ [x_{i,j} = \text{sign}(w_j) \implies x_{m,j} = x_{i,j} = \text{sign}(w_j)] \tag{12}$$

From (12) we know that $\{j \in \mathcal{J} | x_{i,j} = \text{sign}(w_j)\} \subset \{j \in \mathcal{J} | x_{m,j} = \text{sign}(w_j)\}$. So $\{j \in \mathcal{J} | x_{i,j} \neq \text{sign}(w_j)\} \supset \{j \in \mathcal{J} | x_{m,j} \neq \text{sign}(w_j)\}$. The subset cannot be equal because $\mathbf{x}_i \neq \mathbf{x}_m$. And we can split the set $\{j \in \mathcal{J} | x_{m,j} = \text{sign}(w_j)\}$ as the following:

$$\{j \in \mathcal{J} | x_{m,j} = \text{sign}(w_j)\}$$
$$= \{j \in \mathcal{J} | x_{m,j} = \text{sign}(w_j) \land x_{i,j} = \text{sign}(w_j)\} \cup \{j \in \mathcal{J} | x_{m,j} = \text{sign}(w_j) \land x_{i,j} \neq \text{sign}(w_j)\}$$

Figure B.1 helps to demonstrate the relations between the sets.

Now, we compute the raw output of the original conjunctive node with weight $\mathbf{w}$ over negative example $\mathbf{x}_m$:

$$f_{\mathbf{w}}(\mathbf{x}_m) = \sum_{j \in \mathcal{J}} w_j x_{m,j} + \beta$$
$$= \sum_{j \in \mathcal{J} : x_j = \text{sign}(w_j)} |w_j| - \sum_{j \in \mathcal{J} : x_j \neq \text{sign}(w_j)} |w_j| + \beta \tag{13}$$

We can rewrite Equation (13) as the following:

$$f_{\mathbf{w}}(\mathbf{x}_m) = \sum_{\substack{j \in \mathcal{J}: \\ x_{m,j}=\text{sign}(w_j), \\ x_{i,j}=\text{sign}(w_j)}} |w_j| + \sum_{\substack{j \in \mathcal{J}: \\ x_{m,j}=\text{sign}(w_j), \\ x_{i,j}\neq\text{sign}(w_j)}} |w_j| - \sum_{\substack{j \in \mathcal{J}: \\ x_{m,j}\neq\text{sign}(w_j), \\ x_{i,j}\neq\text{sign}(w_j)}} |w_j| + \beta \tag{14}$$

If we flip the sign of the second term on the R.H.S. of Equation (14), since it is guaranteed to be greater than 0, the following inequality holds:

$$f_{\mathbf{w}}(\mathbf{x}) > \sum_{\substack{j \in \mathcal{J}: \\ x_{m,j}=\text{sign}(w_j), \\ x_{i,j}=\text{sign}(w_j)}} |w_j| - \sum_{\substack{j \in \mathcal{J}: \\ x_{m,j}=\text{sign}(w_j), \\ x_{i,j}\neq\text{sign}(w_j)}} |w_j| - \sum_{\substack{j \in \mathcal{J}: \\ x_{m,j}\neq\text{sign}(w_j), \\ x_{i,j}\neq\text{sign}(w_j)}} |w_j| + \beta \tag{15}$$

Using the illustration in Figure B.1, we can transform the R.H.S of (15):

$$\sum_{j \in \mathcal{J} : x_{i,j}=\text{sign}(w_j)} |w_j| - \sum_{j \in \mathcal{J} : x_{i,j}\neq\text{sign}(w_j)} |w_j| + \beta = f_{\mathbf{w}}(\mathbf{x}_i) > 0. \tag{16}$$

Now we have $f_{\mathbf{w}}(\mathbf{x}_m) > f_{\mathbf{w}}(\mathbf{x}_i) > 0$, but this contradicts our assumption that $\mathbf{x}_m \in \mathcal{X}^-$ such that $f_{\mathbf{w}}(\mathbf{x}_m) \leq 0$. So we just proved that no negative examples will be covered by the set $\{\tilde{\mathbf{w}}_1, \ldots, \tilde{\mathbf{w}}_{|\mathcal{X}^+|}\}$ constructed under Equation (4).

$\blacksquare$

We have shown that the split weight set $\{\tilde{\mathbf{w}}_1, \ldots, \tilde{\mathbf{w}}_{|\mathcal{X}^+|}\}$ constructed under Equation (4) will cover all positive examples and none of the negative examples.

## B.2. Optimisation

### B.2.1. Subsumption

Following the setting in Section 3.2, if $\tilde{\mathbf{w}}_p$ associated to $\mathbf{x}_p$ is subsumed by $\tilde{\mathbf{w}}_q$, $\tilde{\mathbf{w}}_q$ also covers the positive example $\mathbf{x}_p$.

**Proof** We need to show that the raw output of $\tilde{\mathbf{w}}_q$ with input $\mathbf{x}_p$ is 6 to prove its coverage.

$$
\begin{aligned}
&\sum_{j \in \mathcal{J}} \tilde{w}_{q,j} x_{p,j} + \left( 6 - \sum_{j \in \mathcal{J}} |\tilde{w}_{q,j}| \right) \\
&= 6 \sum_{j \in \mathcal{J}} \mathbb{1}\left( \tilde{w}_{q,j} = \tilde{w}_{p,j} \right) \tilde{w}_{p,j} x_{p,j} + \\
&\quad \left( 6 - 6 \sum_{j \in \mathcal{J}} \mathbb{1}\left( \tilde{w}_{q,j} = \tilde{w}_{p,j} \right) |\tilde{w}_{p,j}| \right) \quad\quad \text{(by def. of subsumption)} \\
&= 6 \sum_{j \in \mathcal{J}} \mathbb{1}\left( \tilde{w}_{q,j} = \tilde{w}_{p,j} \right) \mathbb{1}\left( x_{p,j} = \operatorname{sign}(w_j) \right) + \\
&\quad \left( 6 - 6 \sum_{j \in \mathcal{J}} \mathbb{1}\left( \tilde{w}_{q,j} = \tilde{w}_{p,j} \right) \mathbb{1}\left( x_{p,j} = \operatorname{sign}(w_j) \right) \right) \quad \text{(by Equation (4))} \\
&= 6
\end{aligned}
\tag{17}
$$

$\blacksquare$

### B.2.2. Remarks for Computing Smaller Split Weight Set

**Remark 3** *Given a sample $\mathbf{x}^{(k)} \in \{-1, 1\}^s$, we can split $\mathcal{J}$ into two sets, based on whether the signs of $w_j$ and $x_j^{(k)}$ are matching:*

$$
\begin{aligned}
\mathcal{J}^{(k)+} &= \left\{ j \in \mathcal{J} \ \middle| \ w_j x_j^{(k)} > 0 \right\} \\
\mathcal{J}^{(k)-} &= \left\{ j \in \mathcal{J} \ \middle| \ w_j x_j^{(k)} < 0 \right\} \\
\mathcal{J}^{(k)+} &\cup \mathcal{J}^{(k)-} = \mathcal{J}
\end{aligned}
\tag{18}
$$

*We can rewrite Equation (2) as: $f_{\mathbf{w}}\left(\mathbf{x}^{(k)}\right) = \max_{j \in \mathcal{J}} |w_j| - 2 \sum_{j \in \mathcal{J}^{(k)-}} |w_j|$. Given $\mathbf{x}^{(k)}$, the bivalent interpretation of this node $b^{(k)} = \top$ when $f_{\mathbf{w}}(\mathbf{x}^{(k)}) > 0$. In the context of*

*splitting a positively used conjunctive node, we care about the following property:*

$$f_{\mathbf{w}}(\mathbf{x}^{(k)}) > 0 \iff \max_{j \in \mathcal{J}} |w_j| > 2 \sum_{j \in \mathcal{J}^{(k)-}} |w_j| \tag{19}$$

Recall the translated rule set $\mathfrak{L}$ that covers the soft-valued truth table of the original conjunctive node, indexed by $\ell$, defined in Equation (5). Following the splitting method described in Section 3.1, at least one of the rules in $\mathfrak{L}$ will be triggered for any positive sample $\mathbf{x}^{(n)}$. If Rule $\ell$ is triggered by $\mathbf{x}^{(n)}$, i.e. $f_{\tilde{\mathbf{w}}_\ell}(\mathbf{x}^{(n)}) > 0$. Using Remark 3, we know that for any $j \in \mathcal{J}$ s.t. $w_j \geq 1/2 \max_{j \in \mathcal{J}} |w_j|$, $j \notin \mathcal{J}^{(n)-}$ and $j \in \mathcal{J}_\ell^+$ for all $\ell$. The intuition is that every large positive weight should be in all rules' positive sets of atoms $\mathcal{J}_\ell^+$, since the large weight indicates a heavy contribution from the input to the output bivalent interpretation. Similarly, if $w_j \leq -1/2 \max_{j \in \mathcal{J}} |w_j|$, then $j \in \mathcal{J}_\ell^-$ for all $\ell$. This property has been discussed in Section 3.2, and we can formalise it as:

**Remark 4** *We split the relevant set $\mathcal{J}$ into $\mathcal{J}^+ = \{j \in \mathcal{J}|w_j > 0\}$ and $\mathcal{J}^- = \{j \in \mathcal{J}|w_j < 0\}$, to separate the positive and negative weights. We define the following sets of indices:*

$$\bar{\mathcal{J}} = \left\{ j \in \mathcal{J} \;\middle|\; |w_j| < \frac{1}{2} \max_{j' \in \mathcal{J}} |w_{j'}| \right\}$$
$$\bar{\mathcal{J}}^+ = \bar{\mathcal{J}} \cap \mathcal{J}^+ \qquad\qquad\qquad \textit{small but positive weights}$$
$$\bar{\mathcal{J}}^- = \bar{\mathcal{J}} \cap \mathcal{J}^- \qquad\qquad\qquad \textit{small but negative weights}$$

*The indices where the weights' absolute values are greater than half of the maximum absolute weight should be included in all split rules:*

$$\forall \ell.j. \left[ j \in (\mathcal{J}^+ \setminus \bar{\mathcal{J}}^+) \implies j \in \mathcal{J}_\ell^+ \right]$$
$$\forall \ell.j. \left[ j \in (\mathcal{J}^- \setminus \bar{\mathcal{J}}^-) \implies j \in \mathcal{J}_\ell^- \right]$$

### B.3. Pseudocode

Algorithm 1 shows the full pseudocode on how to split a positively used conjunctive node with optimisation.

---

**Algorithm 1:** The full algorithm on splitting a positively used conjunctive node with optimisation

---

**Input:** Weight $\mathbf{w}$ of a conjunctive node positively connected to a disjunctive node
**Output:** List of valid split weights

**1** Compute $\mathcal{J} \leftarrow \{j \in \{1..s\} | w_j \neq 0\}$
**2** Compute $\bar{\mathcal{J}} \leftarrow \{j \in \mathcal{J} \mid |w_j| < \max |\mathbf{w}|/2\}$ (Remark 4)
**3 if** $\bar{\mathcal{J}} = \emptyset$ **then return** $[sign(\mathbf{w}) \cdot 6]$ ;
    // Use breadth-first search to explore subsets of $\bar{\mathcal{J}}$ that valid
    // exclusion candidates $\mathcal{E}$ while maximising its cardinality, where:
    // $\sum_{j \in \mathcal{E}} |w_j| < \max |\mathbf{w}|/2$
**4** $valid\_splits \leftarrow \{\}, queue \leftarrow$ empty OrderedDict ;
**5** Add all $j \in \bar{\mathcal{J}}$ as a set $\{j\}$ to $queue$ with parent = None ;
**6 forall** $j \in \bar{\mathcal{J}}$ **do** $queue.add(\{j\}, \text{parent} = \text{None})$ ;
**7 while** *queue is not empty* **do**
**8**    Get the head item $(removal\_indices, parent)$ from $queue$ ;
**9**    $new\_half\_max \leftarrow \max |\mathbf{w}|/2 - \sum |\mathbf{w}[removal\_indices]|$;
**10**    $\mathcal{J}_{new} \leftarrow \{j \in \mathcal{J} \mid j \notin removal\_indices \text{ and } |w_j| < new\_half\_max\}$ ;
**11**    **if** $new\_half\_max \leq 0$ **or** $\mathcal{J}_{new} = \emptyset$ **then**
        // The current 'removal_indices' is a not valid exclusion set or
        // there are no more elements in $\hat{\mathcal{J}}$ that can be added to the
        // exclusion set.
**12**        **if** $new\_half\_max > 0$ **then** $valid\_splits.add(removal\_indices)$ ;
**13**        **else if** *parent is not None* **then** $valid\_splits.add(parent)$ ;
**14**        **continue**;
    // Keep exploring bigger exclusion sets
**15**    $\mathcal{J}_{new} \leftarrow \mathcal{J}_{new} - removal\_indices$ ;
**16**    **if** $\mathcal{J}_{new}$ *is empty* **then**
**17**        $valid\_splits.add(removal\_indices)$ ;
**18**        **continue**;
**19**    **foreach** $j \in \mathcal{J}_{new}$ **do**
**20**        $new\_removal \leftarrow \text{sort}(removal\_indices \cup \{j\})$ ;
**21**        **if** $new\_removal \notin queue$ **then**
**22**            $queue.add(new\_removal, \text{parent} = removal\_indices)$ ;
**23**    **end**
**24 end**
**25** $split\_tensors \leftarrow []$
**26 foreach** *valid exclusion set* $\mathcal{E}$ **do**
**27**    Create new tensor $\tilde{\mathbf{w}}$ where:
**28**        - $j \in \mathcal{E}$: $\tilde{w}_j = 0$
**29**        - $j \in \mathcal{J} \setminus \mathcal{E}$: $\tilde{w}_j = 6 \cdot \text{sign}(w_j)$
**30**    Add $\tilde{\mathbf{w}}$ to split tensors;
**31 end**
**32 return** $split\_tensors$;

---

## Appendix C. Disentanglement of Negatively Used Conjunction

Now we consider a pruned conjunctive node parameterised with weight $\mathbf{w}$ with a total of $s$ inputs, but it's connected to a disjunctive node with a negative weight. This means that the disjunctive node will fire only when the conjunction is not true.

We reuse the definition of relevant set $\mathcal{J}$ and the two sets of inputs based on the soft-valued truth table $\mathbb{X}^+$ and $\mathbb{X}^-$. We want a set of split weights $\{\tilde{\mathbf{w}}_1, \ldots\}$ with $\tilde{\mathbf{w}}_i \in \{-6, 0, 6\}^d$ such that:

$$\forall \mathbf{x} \in \mathcal{X}^-.\exists \ell \left[ \sum_{j \in \mathcal{J}} \tilde{w}_{\ell,j} x_j + \left( 6 - \sum_{j \in \mathcal{J}} |\tilde{w}_{\ell,j}| \right) = 6 \right] \tag{20}$$

$$\forall \mathbf{x} \in \mathcal{X}^+.\neg \exists \ell \left[ \sum_{j \in \mathcal{J}} \tilde{w}_{\ell,j} x_j + \left( 6 - \sum_{j \in \mathcal{J}} |\tilde{w}_{\ell,j}| \right) > -6 \right] \tag{21}$$

Note that we reverse the condition compared to the positive case in Section 3. If an input results in a less-or-equal-to-0 output for conjunctive node with $\mathbf{w}$, the connected disjunctive node will fire, and we want a new conjunctive node that *fires* under this input and connect it to the same disjunctive node but *positively* (Equation (20)). And the new conjunctive node should *not* cover any of the *positive* examples (Equation (21)).

Here we purpose a valid way of constructing such split weights. For each negative example $\mathbf{x}_i \in \mathcal{X}^-$ (which we do want to cover with the new conjunctive nodes), we associate a split weight $\tilde{\mathbf{w}}_i$ to it, where:

$$\forall j \in \{1 \ldots s\}.\tilde{w}_{i,j} = x_{i,j} \cdot \mathbb{1}\left(x_{i,j} \neq \text{sign}(w_j)\right) \cdot 6 = \begin{cases} 6 \cdot x_{i,j} & \text{if } x_{i,j} \neq \text{sign}(w_j) \\ 0 & \text{otherwise} \end{cases} \tag{22}$$

Note that each element $\tilde{w}_i$ will still be in the set of $\{-6, 0, 6\}$. Doing so we create a set of split weights $\mathcal{W} = \{\tilde{\mathbf{w}}_1, \ldots, \tilde{\mathbf{w}}_{|\mathcal{X}^-|}\}$.

### C.1. Negative Coverage: Negative Examples Fire the Split Nodes

Take a negative example $\mathbf{x}_i \in \mathcal{X}^-$. By construction described in Equation (22), we have:

$$
\begin{aligned}
&\sum_{j \in \mathcal{J}} \tilde{w}_{i,j} x_{i,j} + \left( 6 - \sum_{j \in \mathcal{J}} |\tilde{w}_{i,j}| \right) \\
&= 6 \sum_{j \in \mathcal{J}} \mathbb{1}\left(x_{i,j} \neq \text{sign}(w_j)\right) + \left( 6 - 6 \sum_{j \in \mathcal{J}} \mathbb{1}\left(x_{i,j} \neq \text{sign}(w_j)\right) \right) \\
&= 6
\end{aligned}
\tag{23}
$$

Thus, the negative example $\mathbf{x}_i$ will fire the new conjunction node parameterised by $\tilde{\mathbf{w}}_i$.

Similarly, we can prove that if $\tilde{\mathbf{w}}_p$ subsumes $\tilde{\mathbf{w}}_q$, it also fires with the input of negative example $\mathbf{x}_q$ that is associated to $\tilde{\mathbf{w}}_q$, and $\tilde{\mathbf{w}}_q$ becomes not necessary in the final program. So we can compute a smaller final program: $\{\tilde{\mathbf{w}}_i\}_{i \in \mathcal{I}}$, where $\mathcal{I} = \{i \in \{1..|\mathcal{X}^-|\} \mid \tilde{\mathbf{w}}_i$ is not subsumed by another rule$\}$.

## C.2. Positive Coverage: Positive Examples Do Not Fire Split Nodes

Now we prove that $\mathcal{W}$ does not fire for any of the positive examples, by contradiction.

Assume that the split weight $\tilde{\mathbf{w}}_i$ associated to negative example $\mathbf{x}_i \in \mathcal{X}^-$ is fired for $\mathbf{x}_m \in \mathcal{X}^+$. Then the raw output of $\tilde{\mathbf{w}}_i$ over $\mathbf{x}_m$ should be 6 to cover $\mathbf{x}_m$:

$$f_{\tilde{\mathbf{w}}}(\mathbf{x}_m) = 6$$

$$\sum_{j \in \mathcal{J}} \tilde{w}_{i,j} x_{m,j} + \left(6 - \sum_{j \in \mathcal{J}} |\tilde{w}_{i,j}|\right) = 6$$

$$\Rightarrow \sum_{j \in \mathcal{J}} \tilde{w}_{i,j} x_{m,j} = \sum_{j \in \mathcal{J}} |\tilde{w}_{i,j}| \tag{24}$$

Substitute the definition of $\tilde{w}_{i,j}$ in Equation (22) into Equation (24), and we have:

$$\sum_{j \in \mathcal{J}} x_{i,j} \cdot \mathbb{1}\left(x_{i,j} \neq \mathrm{sign}(w_j)\right) \cdot x_{m,j} = \sum_{j \in \mathcal{J}} \mathbb{1}\left(x_{i,j} \neq \mathrm{sign}(w_j)\right) \tag{25}$$

For Equation (25) to hold, the following has to hold:

$$\forall j \in \mathcal{J}. \; [x_{m,j} = x_{i,j} \implies x_{i,j} \neq \mathrm{sign}(w_j)] \tag{26}$$

From (26) we know that $\{j \in \mathcal{J} | x_{i,j} \neq \mathrm{sign}(w_j)\} \subset \{j \in \mathcal{J} | x_{m,j} \neq \mathrm{sign}(w_j)\}$. So $\{j \in \mathcal{J} | x_{m,j} = \mathrm{sign}(w_j)\} \supset \{j \in \mathcal{J} | x_{i,j} = \mathrm{sign}(w_j)\}$, which means that $x_{m,j} = \mathrm{sign}(w_j)$ holds for any $j \in \mathcal{J}$ such that $x_{i,j} = \mathrm{sign}(w_j)$. The subset cannot be equal because $\mathbf{x}_i \neq \mathbf{x}_m$. And we can split the set $\{j \in \mathcal{J} | x_{m,j} \neq \mathrm{sign}(w_j)\}$ like this:

$$\{j \in \mathcal{J} | x_{m,j} \neq \mathrm{sign}(w_j)\}$$
$$= \{j \in \mathcal{J} | x_{m,j} \neq \mathrm{sign}(w_j) \wedge x_{i,j} \neq \mathrm{sign}(w_j)\} \cup \{j \in \mathcal{J} | x_{m,j} \neq \mathrm{sign}(w_j) \wedge x_{i,j} \neq \mathrm{sign}(w_j)\}$$

Figure C.2 helps to demonstrate the relations between the sets.

$$\{j \in \mathcal{J} | x_{m,j} \neq \mathrm{sign}(w_j) \wedge x_{i,j} = \mathrm{sign}(w_j)\}$$

Now, we compute the raw output of the original conjunctive node with weight $\mathbf{w}$ over positive example $\mathbf{x}_m$:

$$f_{\mathbf{w}}(\mathbf{x}_m) = \sum_{j \in \mathcal{J}} w_j x_{m,j} + \beta$$

$$= \sum_{j \in \mathcal{J} : x_{m,j} = \mathrm{sign}(w_j)} |w_j| - \sum_{j \in \mathcal{J} : x_{m,j} \neq \mathrm{sign}(w_j)} |w_j| + \beta \tag{27}$$

We can rewrite Equation (27) as the following:

$$f_{\mathbf{w}}(\mathbf{x}_m) = \sum_{\substack{j \in \mathcal{J}: \\ x_{m,j}=\mathrm{sign}(w_j), \\ x_{i,j}=\mathrm{sign}(w_j)}} |w_j| \; - \sum_{\substack{j \in \mathcal{J}: \\ x_{m,j}\neq\mathrm{sign}(w_j), \\ x_{i,j}=\mathrm{sign}(w_j)}} |w_j| \; - \sum_{\substack{j \in \mathcal{J}: \\ x_{m,j}\neq\mathrm{sign}(w_j), \\ x_{i,j}\neq\mathrm{sign}(w_j)}} |w_j| \; + \beta \qquad (28)$$

If we flip the sign of the second term in Equation (28), which is guaranteed to be greater than 0, the following inequality holds:

$$f_{\mathbf{w}}(\mathbf{x}_m) < \sum_{\substack{j \in \mathcal{J}: \\ x_{m,j}=\mathrm{sign}(w_j), \\ x_{i,j}=\mathrm{sign}(w_j)}} |w_j| \; + \sum_{\substack{j \in \mathcal{J}: \\ x_{m,j}\neq\mathrm{sign}(w_j), \\ x_{i,j}=\mathrm{sign}(w_j)}} |w_j| \; - \sum_{\substack{j \in \mathcal{J}: \\ x_{m,j}\neq\mathrm{sign}(w_j), \\ x_{i,j}\neq\mathrm{sign}(w_j)}} |w_j| \; + \beta \qquad (29)$$

We can see that the R.H.S of Inequality (29) is the same as:

$$\sum_{j \in \mathcal{J}:x_{i,j}=\mathrm{sign}(w_j)} |w_j| \; - \sum_{j \in \mathcal{J}:x_{i,j}\neq\mathrm{sign}(w_j)} |w_j| \; + \beta$$

which is exactly $f_{\mathbf{w}}(\mathbf{x}_i)$. Because $\mathbf{x}_i \in \mathcal{X}^-$, $f_{\mathbf{w}}(\mathbf{x}_i) < 0$.

Now we have $f_{\mathbf{w}}(\mathbf{x}_m) < f_{\mathbf{w}}(\mathbf{x}_i) < 0$, but this contradicts our assumption that $\mathbf{x}_m \in \mathcal{X}^+$ where $f_{\mathbf{w}}(\mathbf{x}_m) > 0$. So we prove that no positive examples will fire any split node constructed under Equation (22).

We have shown that $\{\tilde{\mathbf{w}}_1, \ldots, \tilde{\mathbf{w}}_{|\mathcal{X}^-|}\}$ constructed under Equation (22) will fire for all negative examples as input and not for any of the positive examples as input. And by pruning split nodes that are subsumed by other split nodes, we can also compute an equally valid but smaller set of split weights $\{\tilde{\mathbf{w}}_1, \ldots, \tilde{\mathbf{w}}_{|\mathcal{Q}|}\}$ where $\mathcal{Q} = \{q \in \{1..|\mathcal{X}^-|\} \mid \tilde{\mathbf{w}}_q$ is not subsumed by another rule$\}$. ∎

## Appendix D. Experiments

### D.1. Interpretability of Models

While many models offer varying degrees of interpretability, decision trees and neural DNF-based models stand out for their high interpretability by providing conditional prediction in a rule-based decision-making process: decision trees provide decision paths that can be followed from inputs to decisions; and the neural DNF-based models are designed such that they provide rule-based interpretations. Logistic regression and random forest are two less interpretable models with composition nature: logistic regression shows the additive contribution of inputs to the log-odds of the final decision; each tree in a random forest is rule-based but the overall decision is derived from a weighted voting system of the trees' decisions. Support Vector Machines (SVMs) and Multi-Layer Perceptrons (MLPs) are typically not considered interpretable due to their complex decision boundaries and lack of transparency in decision-making processes. Because of the different characteristics, we focus on decision trees as direct competitors to neural DNF-based models for our interpretability study, and ignore the other models with non-rule-based interpretations.

## D.2. Experiment Settings

Table 5 shows the experiment settings across the datasets for MLP and neural DNF-based models. For logistic regressions, SVMs, random forests and decision trees, we run the experiments with 5-fold cross validation.

Table 5: Experiment settings for MLP and neural DNF-based models. Predicate invention requirement is only applicable to neural DNF-based models.

| Dataset | Neural DNF-based Models: Requires Predicate Invention? | Hold-out Test / Cross Validaiton | Number of Runs / Folds |
|---|---|---|---|
| **Binary** | | | |
| Monk | No | Hold-out Test | 18 |
| BCC | Yes | Cross Validation | 5 |
| Mushroom | No | Hold-out Test | 16 |
| CDC | Yes | Hold-out Test | 10 |
| **Multiclass** | | | |
| Car | No | Hold-out Test | 16 |
| Covertype | Yes | Hold-out Test | 10 |
| **Multilabel** | | | |
| ARA | No | Cross Validation | 10 |
| Budding | No | Cross Validation | 10 |
| Fission | No | Cross Validation | 10 |
| MAM | No | Cross Validation | 10 |

## D.3. Examples of Logical Interpretation

### D.3.1. Binary Classification

Below is an example of the logical interpretation of a neural DNF model in the the dataset **Monk** (Wnek, 1993).

```
% t is the target class predicate.
% a_i is the input feature.
t :- a_1, a_4.   t :- a_2, a_5.
t :- a_0, a_3.
t :- not a_11, not a_12, not a_13.
```

Below is an example of the logical interpretation of a neural DNF model in the dataset **BCC** (Patrício et al., 2018), where threshold-learning predicate invention method is used.

```
% Invented predicates defined by the threshold-learning
% predicate invention method.
a_2 = feature_0 > -269.8374328613281
a_3 = feature_0 > 186.82131958007812
a_6 = feature_1 > 58.69133377075195
```

```
a_6 = feature_1 > 58.69133377075195
a_8 = feature_2 > 197.28598022460938
a_14 = feature_3 > -13.131363868713379
a_28 = feature_7 > 4.719961166381836

% Rules
% t is the target class predicate.
% a_i are invented predicates shown above.
t :- a_2, not a_3, not a_6.
t :- a_8, a_14.
t :- a_2, not a_6, a_28.
```

Below is an example of the logical interpretation of a neural DNF model in the dataset **Mushroom** (Guide, 1981).

```
% t is the target class predicate.
% a_i are input features.
t :- a_50, a_97, a_103.
t :- a_46, a_89, not a_105, not a_109.
t :- not a_21, not a_24, not a_26.
```

Below is an example of the logical interpretation of a neural DNF model in the dataset **CDC** diabetes (Teboul, 2021), where threshold-learning predicate invention method is used.

```
% Invented predicates defined by the threshold-learning
% predicate invention method.
a_2 = feature_0 > 74.53063201904297
a_5 = feature_0 > 58.61220932006836
a_6 = feature_0 > -33.472694396972656
a_7 = feature_0 > 102.41500854492188
a_8 = feature_1 > 22.021968841552734
a_12 = feature_1 > 16.420833587646484
a_15 = feature_1 > 19.272249221801758
a_18 = feature_2 > 45.10814666748047
a_25 = feature_3 > 51.938621520996094

% Rules
% t is the target class predicate.
% a_i with i in [0, 63] are invented predicates shown above.
% a_i with i in [63, 91] are binary input features.
t :- a_35.
t :- a_38.
t :- a_52.
t :- a_47.
t :- a_43.
t :- not a_37.
t :- a_39.
t :- a_45.
t :- a_7.
t :- a_25.
t :- a_18.
t :- a_2, a_12, not a_32, not a_41.
t :- a_2, a_15, not a_32.
t :- not a_34, a_36, not a_42, not a_59.
t :- a_8, a_36, not a_42, not a_59.
t :- a_36, not a_42, a_55.
```

```
t :- a_48.
t :- not a_41, not a_42, not a_59.
t :- not a_34, a_36, not a_51.
t :- not a_49, a_53.
t :- a_5, not a_42, not a_50, not a_51, not a_57.
t :- not a_6, not a_59.
t :- a_40, not a_51.
t :- a_36, a_53.
```

### D.3.2. MULTICLASS CLASSIFICATION

Below is an example of the logical interpretation of a neural DNF-MT model in the dataset
**Car** (Bohanec, 1988).

```
% conj_i are the output predicates of the conjunctive layer.
% a_i are the input features.
conj_0 :- not a_13, not a_14.
conj_1 :- a_2, not a_5, not a_12, not a_17, a_20.
conj_1 :- a_2, not a_12, a_18.
conj_1 :- not a_7, not a_12, a_18.
conj_1 :- a_1, not a_12, a_18.
conj_1 :- a_1, not a_5, not a_16, a_20.
conj_3 :- not a_0, not a_3, a_5, not a_8, a_14, not a_17,
    a_20.
conj_3 :- a_1, a_5, not a_9, not a_12, not a_17, a_20.
conj_6 :- not a_9, not a_10, not a_13, a_17.
conj_7 :- not a_2, not a_5, a_9, not a_14, a_16, not a_18.
conj_7 :- not a_1, not a_6, not a_11, not a_14, a_16,
    not a_18.
conj_8 :- a_2, a_5, not a_12, not a_19, not a_20.
conj_8 :- a_2, a_5, a_13, not a_19.
conj_9 :- not a_7, not a_17, not a_19.
conj_9 :- not a_3, not a_7, not a_19.
conj_9 :- not a_0, not a_3, not a_19, not a_20.
conj_12 :- not a_20.
conj_13 :- a_3, not a_5, not a_6.
conj_14 :- a_1, a_5, not a_12, not a_19, not a_20.
conj_15 :- a_1, not a_7, not a_12, not a_17, a_18.
conj_16 :- not a_0.
conj_17 :- a_0, not a_7, not a_12, not a_17, not a_19.
conj_21 :- a_1, a_6, not a_12, not a_19.
conj_22 :- not a_0, not a_3, not a_4, not a_7, not a_8,
    not a_12, not a_17, not a_19, not a_20.
conj_24 :- not a_5, not a_6, a_17, a_20.
% During inference, given an input, we first compute the
% conjunctions that are activated.
% Then we generate the ProbLog rule.
% For example:
% Input (tensor):
[-1., -1.,  1., -1., -1., -1., -1.,  1., -1.,  1., -1.,
-1., -1., -1.,  1., -1.,  1., -1., -1., -1.,  1.]
% Input (translated to predicates):
[a_2, a_7, a_9, a_14, a_16, a_20]
% Conjunctions that are activated:
[conj_1, conj_16]
```

```
% ProbLog rule:
0.806::class_0 ; 0.020::class_1 ; 0.171::class_2 ;
0.003::class_3 :- conj_1, conj_16.
% Final prediction: class_0, ground truth: class_0
```

Below is an example of the logical interpretation of a neural DNF-MT model in the dataset **Covertype** (Blackard, 1998), where an MLP is used as a predicate inventor.

```
% conj_i are the output predicates of the conjunctive layer.
% a_i are the invented predicates from the MLP predicate
% inventor.
conj_0 :- not a_14.
conj_1 :- not a_6, a_27, not a_57.
conj_2 :- a_1.
conj_3 :- not a_46, a_52.
conj_6 :- not a_26.
conj_7 :- a_42.
conj_10 :- a_24.
conj_13 :- a_0.
conj_16 :- a_12, a_36, not a_51.
conj_17 :- not a_3.
conj_18 :- not a_50, a_52.
conj_19 :- not a_18, a_62.
conj_19 :- a_34.
conj_20 :- not a_57.
conj_21 :- a_12, a_17.
conj_22 :- a_0, not a_41.
conj_23 :- not a_49, not a_61.
conj_24 :- a_20.
conj_25 :- not a_21, not a_26.
conj_26 :- a_63.
conj_27 :- not a_51, a_56.
conj_28 :- not a_26, a_39.
conj_29 :- a_39.
conj_30 :- not a_26, a_48.
conj_32 :- not a_40.
conj_33 :- a_44.
conj_34 :- a_10.
conj_35 :- not a_44.
conj_36 :- a_33.
conj_37 :- not a_42.
conj_38 :- not a_44.
conj_40 :- a_29.
conj_41 :- not a_55.
conj_42 :- not a_10, not a_58.
conj_43 :- not a_61, a_62.
conj_44 :- not a_48.
conj_45 :- a_41, not a_50.
conj_46 :- a_31.
conj_47 :- a_43.
conj_48 :- a_12.
conj_50 :- not a_54.
conj_51 :- a_19, not a_59.
conj_52 :- not a_1, not a_2, a_9, not a_51.
conj_53 :- not a_15, not a_22, not a_29, not a_38, not a_43.
conj_54 :- a_9, not a_15, a_37.
```

```
conj_55 :- a_49.
conj_56 :- not a_49, not a_53.
conj_57 :- a_36, not a_37.

% During inference, given an input, we first compute the
% invented predicates from the MLP predicate inventor.
% Then we check which conj_i are activated.
% Finally, we generate the ProbLog rule.
% For example:
% Input:
[ 1.3166, -0.5331, -0.2809, -0.1949, -0.4017,  0.2064,  1.0031,
-0.0667, -0.2774,  1.0000, -1.0000,  1.0000,  1.0000,  1.0000,
-1.0000, -1.0000]
% Invented predicates that are true:
[a_0, a_1, a_2, a_3, a_7, a_8, a_9, a_10, a_11, a_14, a_15, a_17,
a_22, a_26, a_27, a_28, a_29, a_31, a_32, a_33, a_34, a_38, a_39,
a_41, a_43, a_44, a_47, a_48, a_49, a_51, a_53, a_54, a_58, a_59,
a_60, a_61, a_63]
% Conjunctions that are activated:
[conj_1, conj_2, conj_13, conj_19, conj_20, conj_26, conj_29,
conj_32, conj_33, conj_34, conj_36, conj_37, conj_40, conj_41,
conj_45, conj_46, conj_47, conj_55]
% ProbLog rule:
0.005::class_0 ; 0.047::class_1 ; 0.000::class_2 ;
0.000::class_3 ; 0.128::class_4 ; 0.001::class_5 ;
0.820::class_6 :- conj_1, conj_2, conj_13, conj_19, conj_20,
conj_26, conj_29, conj_32, conj_33, conj_34, conj_36, conj_37,
conj_40, conj_41, conj_45, conj_46, conj_47, conj_55.
% Final prediction: class_6, ground truth: class_6
```

### D.3.3. Multilabel Classification

The datasets in this section are Boolean Network (Kauffman, 1969) datasets, with ground-truth logic programs. Boolean Network models the state transitions of genes from time $t$ to $t + 1$, and we treat the learning of such a dynamic system as a multilabel classification problem: the inputs are gene states at time $t$ and the labels are the gene states at time $t + 1$.

Below is an example of the logical interpretation of a neural DNF model in the dataset **ARA** (Chaos et al., 2006).

```
% l_i are the target labels.
% a_i are inputs.
l_0 :- a_1, a_6.
l_0 :- a_0, a_4, a_13, a_14.
l_0 :- a_0, a_9, a_13, a_14.
l_1 :- a_1.
l_2 :- not a_4, not a_12.
l_3 :- not a_5.
l_4 :- a_6, not a_9.
l_4 :- a_3, not a_9.
l_4 :- not a_9, not a_12.
l_5 :- not a_6.
l_6 :- not a_12.
```

```
l_6 :- not a_5.
l_7 :- not a_12.
l_8 :- a_8, not a_9.
l_8 :- a_8, not a_14.
l_9 :- not a_7, not a_12.
l_9 :- a_6, a_8.
l_9 :- a_6, not a_11.
l_9 :- not a_4, a_6.
l_9 :- a_6, not a_10.
l_12 :- not a_4, a_5, not a_6.
l_13 :- a_6, a_9.
l_13 :- a_0, a_6.
l_14 :- a_6.
```

Below is an example of the logical interpretation of a neural DNF model in the dataset **Budding** (Li et al., 2004).

```
% l_i are the target labels.
% a_i are inputs.
l_1 :- a_0.
l_2 :- a_2, not a_8.
l_2 :- a_1, not a_8.
l_2 :- a_1, a_2.
l_3 :- a_3, not a_8.
l_3 :- a_1, a_3.
l_3 :- a_1, not a_8.
l_4 :- a_2.
l_5 :- not a_4, not a_8, not a_10, a_11..
l_5 :- not a_4, not a_6, not a_8, a_10.
l_5 :- not a_4, not a_6, a_10, a_11.
l_5 :- not a_6, not a_8, a_10, a_11.
l_5 :- a_5, not a_6, not a_8, a_11.
l_5 :- not a_4, a_5, not a_6, not a_8.
l_5 :- not a_4, a_5, not a_8, a_10.
l_5 :- not a_4, a_5, a_10, a_11.
l_5 :- a_5, not a_8, a_10, a_11.
l_5 :- not a_4, a_5, not a_6, a_11.
l_5 :- a_5, not a_6, a_10, a_11.
l_5 :- a_5, not a_6, not a_8, a_10.
l_5 :- not a_4, a_5, not a_6, a_10.
l_6 :- a_3, not a_5, not a_10.
l_6 :- not a_5, a_6, not a_10.
l_6 :- a_3, a_6, not a_10.
l_6 :- a_3, not a_5, a_6.
l_7 :- not a_4, a_7, not a_8, a_10.
l_7 :- not a_4, not a_6, not a_8, a_10.
l_7 :- not a_4, not a_6, a_7, a_10.
l_7 :- not a_4, not a_6, a_7, not a_8.
l_7 :- not a_6, a_7, not a_8, a_10.
l_8 :- not a_5, a_6, not a_7, a_8.
l_8 :- a_6, not a_7, a_9, not a_10.
l_8 :- not a_5, not a_7, a_9, not a_10.
l_8 :- not a_5, a_6, not a_7, a_9.
l_8 :- a_6, not a_7, a_8, a_9.
l_8 :- not a_5, a_6, not a_7, not a_10.
l_8 :- not a_7, a_8, a_9, not a_10.
```

```
l_8 :- not a_5, not a_7, a_8, a_9.
l_8 :- a_6, not a_7, a_8, not a_10.
l_8 :- not a_5, not a_7, a_8, not a_10.
l_8 :- not a_5, a_8, a_9, not a_10.
l_8 :- not a_5, a_6, a_9, not a_10.
l_8 :- not a_5, a_6, a_8, not a_10.
l_8 :- not a_5, a_6, a_8, a_9.
l_8 :- a_6, a_8, a_9, not a_10.
l_9 :- a_6.
l_9 :- a_8.
l_10 :- a_8.
l_10 :- a_9.
l_11 :- a_9, a_10.
l_11 :- not a_8, a_9.
l_11 :- not a_8, a_10.
```

Below is an example of the logical interpretation of a neural DNF model in the dataset **Fission** (Davidich and Bornholdt, 2008).

```
% l_i are the target labels.
% a_i are inputs.
l_1 :- a_0.
l_2 :- a_2, not a_3, a_5, not a_9.
l_2 :- not a_1, not a_3, a_5, not a_9.
l_2 :- not a_1, a_2, not a_3, a_5.
l_2 :- not a_1, a_2, not a_3, not a_9.
l_2 :- not a_1, a_2, a_5, not a_9.
l_3 :- not a_2, a_3, not a_4.
l_3 :- a_3, not a_4, not a_7.
l_3 :- not a_2, a_3, not a_7.
l_3 :- not a_2, not a_4, not a_7.
l_4 :- not a_1, a_4, a_5, not a_9.
l_4 :- not a_1, not a_3, not a_9.
l_4 :- not a_1, not a_3, not a_4, not a_5.
l_4 :- not a_3, a_4, a_5, a_9.
l_5 :- a_7.
l_6 :- a_3, not a_5.
l_6 :- not a_5, a_6.
l_6 :- a_3, a_6.
l_7 :- a_9.
l_8 :- not a_3, a_8.
l_8 :- not a_3, a_5.
l_8 :- a_5, a_8.
l_9 :- not a_2, not a_4, a_6, not a_7, not a_8, a_9.
```

Below is an example of the logical interpretation of a neural DNF model in the dataset **MAM** (Fauré et al., 2006).

```
% l_i are the target labels.
% a_i are inputs.
l_0 :- a_0.
l_1 :- not a_2, a_3.
l_2 :- not a_0, a_5, not a_9.
l_2 :- not a_0, not a_1, not a_4, not a_9.
l_3 :- not a_2, a_5, not a_9.
l_3 :- not a_2, not a_4, not a_9.
```

```
l_4 :- not a_2, a_3, not a_6, not a_8.
l_4 :- not a_2, a_3, not a_6, not a_7.
l_4 :- not a_2, a_4, not a_6, not a_8.
l_4 :- not a_2, a_4, not a_6, not a_7.
l_5 :- not a_0, not a_4, a_5, not a_9.
l_5 :- not a_0, not a_1, a_5, not a_9.
l_5 :- not a_0, not a_1, not a_4, not a_9.
l_6 :- a_9.
l_7 :- not a_8.
l_7 :- a_7, a_9.
l_7 :- a_6, a_7.
l_7 :- a_4, a_7.
l_8 :- a_6.
l_8 :- a_5, a_9.
l_8 :- not a_4, not a_9.
l_9 :- not a_6, not a_8
```

