# OpenReview forum: "Disentangling Neural Disjunctive Normal Form Models"
_nesyconf.org/NeSy/2025/Conference_Phase_2 — NeSy 2025 - Phase 2 Poster_

### Official Review · Reviewer_3vkc · 2025-06-30
**Nice idea but a bit undercooked**

**Rating:** 5
**Confidence:** 4

**Review:**

This work addresses translating Neural Disjunctive Normal Form-based models into interpretable logical representation.
Hereby, the presented method aims to disentangle the semi-symbolic nodes, i.e., split them into multiple nodes, to better preserve the original model's performance.
To this end, they employ a discretization scheme based on introducing a soft-valued truth table.

Strengths
- The presented method can improve the interpretation of neural (DNF) models through their translation into logical  formulas.
- Overall, the method itself is presented well and it is easy to follow along. The approach seems plausible and the experimental results are promising.

Weaknesses:
- The authors mention in Section 4, regarding their predicate invention, that a solution is guaranteed in contrast to Baugh et al. (2025). The text does not support this claim very clearly and should be improved. Overall, this section is very short and more challenging to follow than sections 2 and 3.
- The runtime behavior shown in Figure 3 did not become clear. The data is shown without standard deviation and unaddressed behavior (e.g., the dip in runtime at about 29 inputs), undermining the value of the Figure and the discussion of the method's limitations. Similarly, the runtime behavior of the resulting logical model remains unaddressed, e.g., the impact of the translation and disentanglement on inference time. Considering the exponential runtime of the method, it is important to improve on this part of the paper.
- The paper does not address the advantages or exemplify the improved interpretability of the translated neural model well. While the interpretatability of methods such as decision trees is quickly dismissed considering their potential size (page 1), the paper fails to fully support that the presented method's output is competitive in terms of human interpretability and does not fail in the same way due to size and complexity.

Minor:
- p3: The sentence "One discretization method used in previous works is ..." is missing references to such works.
- p3: In the sentence "... convert the weights from a continuous range to a fixed valued range {-6, 0, 6}^N ... " it should be a set, not a range.

Overall, the work heads in the right direction but has several shortcomings which makes me tend more towards rejection.

**Anonymity:**

Remain anonymous

---

### Official Review · Reviewer_dGbT · 2025-07-07
**Disentangling Neural Disjunctive Normal Form Models**

**Rating:** 7
**Confidence:** 5

**Review:**

The paper is well-organized, presenting a logical flow from identifying the challenge of entanglement in neural DNF models to proposing solutions through disentanglement and predicate invention, and concluding with robust experimental validation and comparisons.

Comments:
1. The threshold-learning predicate invention method is described as learning m threshold values for each real-valued feature. However, the paper lacks a detailed explanation on how m is determined or how the temperature parameter T is optimized, leaving some ambiguity in the methodology.

2. While the paper reports F1 scores to demonstrate performance, it omits additional metrics such as precision, recall, or accuracy. Furthermore, it does not address runtime comparisons or the computational costs associated with the disentanglement process, which would provide a more comprehensive evaluation.

3. The paper compares neural DNF models to MLPs, decision trees, and OSDT, noting superior performance in 7 out of 10 datasets. However, it does not offer a detailed analysis of why the proposed model outperforms specific baselines across different task types (binary, multiclass, or multilabel), limiting insight into the architectural or methodological advantages driving these results.

**Anonymity:**

Remain anonymous

---

### Official Review · Reviewer_r4x5 · 2025-07-08
**An improvement over an earlier published approach, learning shorter rules than traditional methods while performance is comparable. Guarantees on performance are unknown, serious limitations remain.**

**Rating:** 6
**Confidence:** 3

**Review:**

The paper is based on earlier work on the neural DNF model. This is a neural network with an architecture that mimics the shape of a DNF formula. A trained neural DNF model can be converted into a symbolic representation, but until now, this conversion process came at the price of a large loss in accuracy. The current paper presents an improved "disentanglement method" which reduces this loss in accuracy during the extraction of the symbolic representation.

The disentanglement method is described fairly clearly, but it remains unclear whether there are any guarantees on the loss in accuracy during the newly proposed distentangelment step.

The value of the paper depends heavily on the value of the DNF model it aims to improve. The method itself is evaluated on a set of rather simply classification tasks which can also be reliably solved by decision tree learning (either optimise sparse trees or plain). The main advantage over these classical methods is not the gain in accuracy, but the reduced length of the learned/extracted symbolic representation.

The discussion section fairly discusses the limitations of the proposed method: a rather strong requirement on node inputs to be discrete, and the method scales poorly (exponentially) with the number of inputs use in a node.

**Anonymity:**

Disclose identity